# Numerical Simulation of Water Absorption and Swelling in Dehulled Barley Grains during Canned Porridge Cooking

**Lei Wang [1], Mengting Wang [1], Mingming Guo [1,2,3], Xingqian Ye [1,2,3], Tian Ding [1,2,3] and Donghong Liu [1,2,3,\*]**

1  College of Biosystems Engineering and Food Science, Zhejiang University, Hangzhou 310058, China; leiwang94@zju.edu.cn (L.W.); mtwang@zju.edu.cn (M.W.); mingguo@zju.edu.cn (M.G.); psu@zju.edu.cn (X.Y.); tding@zju.edu.cn (T.D.)
2  Fuli Institute of Food Science, Zhejiang University, Hangzhou 310058, China
3  Zhejiang Key Laboratory for Agri-Food Processing, National Engineering Laboratory of Intelligent Food Technology and Equipment, Hangzhou 310058, China
\*  Correspondence: dhliu@zju.edu.cn; Tel.: +86-0571-8898-2169; Fax: +86-0571-8898-2169

**Abstract:** Understanding the hydration behavior of cereals during cooking is industrially important in order to optimize processing conditions. In this study, barley porridge was cooked in a sealed tin can at 100, 115, and 121 °C, respectively, and changes in water uptake and hygroscopic swelling in dehulled barley grains were measured during the cooking of canned porridge. In order to describe and better understand the hydration behaviors of barley grains during the cooking process, a three-dimensional (3D) numerical model was developed and validated. The proposed model was found to be adequate for representing the moisture absorption characteristics with a mean relative deviation modulus (P) ranging from 4.325% to 5.058%. The analysis of the 3D simulation of hygroscopic swelling was satisfactory for describing the expansion in the geometry of barley. Given that the model represented the experimental values adequately, it can be applied to the simulation and design of cooking processes of cereals grains, allowing for saving in both time and costs.

**Keywords:** barley; simulation; hydration; swelling; cooking; porridge

## 1. Introduction

Barley (*Hordeum vulgare* L.) is an ancient and widely adapted grain. It ranks fourth among grains in terms of quantity produced (142 M mt, 2014–2017 mean), behind corn (*Zea mays* L., 1027 M mt), wheat (*Triticum aestivum* L., 744 M mt), and rice (*Oryza sativa* L., 482 M mt), and ahead of sorghum (*Sorghum bicolor*, 63 M mt), oat (*Avena sativa*, 23 M mt), and rye (*Secale cereale* L., 13 M mt) [1]. In recent times, because of the high dietary fiber content of barley and the effectiveness of barley β-glucan in lowering cholesterol, the interest in barley food products—such as tea, soup, beverage, snacks, and porridge—is increasing worldwide [2,3].

Barley porridge, similarly to other cereal porridge, is a traditional food in eastern countries. Conventionally, whole or pearled barley grain is boiled in water to gelatinize starch in barley and fully expand it [3], but the cooked barley porridge can only be stored for a few days at ambient temperature. Additionally, ready-to-eat (RTE) porridge has attracted a great deal of attention in many countries due to its excellent storage stability [4]. However, most commercial instant porridge needs to be mixed with hot water before being consumed. Hence, canned barley porridge is more attractive because it could be consumed without any preparation [5].

The cooking of barley porridge in industry is a hydrothermal process to provide the desired attributes in the final product, which are strongly affected by cooking conditions, such as temperature, processing time, and cooking media [6]. Changes in the moisture content and volume of barley grains are the two main phenomena during cooking. Controlling the moisture content in barley grains during cooking is of great importance as water molecules play many roles in food reactions and food quality [7]. In order to predict the optimum processing conditions, the water content variations with the grains are needed as quantitative information [8]. Therefore, it is essential to understand the hydration kinetics of kernels during cooking, and the effect of cooking conditions on the assimilation of moisture [9]. What is more, the swelling of kernels during cooking affects the moisture absorption rate [10]. Hence, the instantaneous kernel volume is a key parameter for better understanding the water absorption process [11]. Many empirical models have been applied to predict the hydration behavior of grains, such as the Expansional, Peleg, and Weibull models. However, considering that these models are just fitted to experimental values, the fitting results are restricted to the test conditions used. Consequently, the phenomenological models are incrementally selected to better describe the phenomena included in the hydration process [12].

Recently, Bakalis et al. [13] used COMSOL Multiphysics®software to simulate the diffusion of moisture in the parboiled grain during cooking. However, their study was limited to starch gelatinization at 70 °C, while cooking is usually done at or above 100 °C. Balbinoti et al. [14] also used COMSOL Multiphysics®software to simulate moisture transfer in the parboiling process step in two- and three-dimensional space under four different temperatures ranging from 35 to 60 °C. Perez et al. [15] built a comparative 3D simulation on water absorption and hygroscopic swelling in japonica rice grains for a soaking temperature of 25, 35, 45, and 55 °C. Montanuci et al. [9] developed a three-dimensional model to describe the hydration process of barley grains under various temperatures, from 10 to 25 °C. To our best knowledge, no research has simulated the hydration curve and large deformation of barley kernels during cooking when the temperature was 100 °C and above.

Therefore, the phenomena of mass transfer, heat transfer, and deformation involved in the cooking period of canned barley porridge are coupled at 100, 115, and 121 °C in this context, in order to evaluate the effect of processing time and temperature on the moisture absorption and volume expansion. Furthermore, the mathematical model developed in this study is also intended to be used for conditions different from those tested in the present study, generating substantial savings in time, energy, and costs by reducing the experimental tests required.

## 2. Model Development

In this section, a numerical model is presented to describe the distribution and amount of moisture content, as well as the volumetric expansion of a dehulled barley grain undergoing hygroscopic swelling at three different temperature cooking conditions. In order to simplify the complexity conditions, the following assumptions are applied: the barley grain was considered as continuous, homogenous and isotropic; the initial moisture content is considered as homogeneous; and the initial surface temperature of the barley grain is equal to the water temperature.

### 2.1. Diffusion

The transient model applied to describe the phenomena of mass transfer and heat transfer during the cooking of canned porridge was developed based on Fick's second Law and Fourier's Law, respectively, according to Equations (1) and (2).

$$\frac{\partial c_i}{\partial t} + \nabla.(-D_i\nabla c_i) + u.\nabla c_i = R_i \tag{1}$$

$$\rho C_p\frac{\partial T}{\partial t} + \rho C_p u\nabla T = \nabla(k\nabla T) + Q \tag{2}$$

where $c_i$, $D_i$ and $u$ are the water concentration (mol/m³), water diffusion coefficient (m²/s), and velocity field (m/s), respectively; $R_i$ is the mass generation (kg/m³), $\rho$ is the barley density (kg/m³), $C_p$ is specific heat (J/kg·K), $k$ is the thermal conductivity (W/m·K), $T$ is the grain temperature (K), and $Q$ is the heat production (W·m).

## 2.2. Hygroscopic Swelling

Dehulled barley grains are predominantly composed of starch (about 65–68%) [2]. During cooking, the starch granules contained in dehulled barley kernel absorb water and swell due to its gelatinization. The hygroscopic strain (i.e., moisture-induced strain) is caused by the swelling of the starch molecules after absorbing water, which can be expressed as an equation in related to the hygroscopic expansion coefficient ($\beta_h$) and the moisture content gradient ($\Delta c$) [16]:

$$\epsilon_{hs} = \beta_h \Delta c \tag{3}$$

## 2.3. Boundary and Initial Conditions

The boundary conditions for the governing equations are shown in Figure 1 and described in detail as below. The initial conditions are also listed in Table 1 with the input parameters.

### 2.3.1. Solid Mechanics

The displacement of side B parallel to the symmetry planes (side C) is set to zero while side A is free to deform (Figure 1).

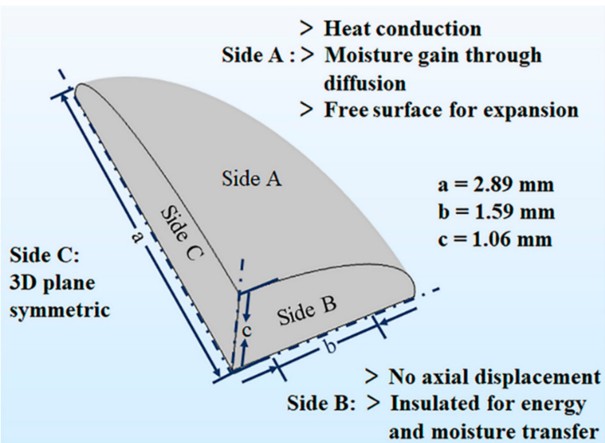

**Figure 1.** Schematic showing the barley geometry used for simulations and boundary conditions for coupled solid mechanics–heat and mass transport model.

### 2.3.2. Heat and Mass Transfer

The initial and boundary conditions established for heat transfer in the barley grain throughout the process are: (a) known initial temperature in the barley grain (Equation (4)); (b) no heat flux across the symmetry region (Equation (5)); and (c) convection occurs at the solid-fluid interface (Equation (6)). For mass transfer, the conditions are: (a) the initial moisture content in the grain is uniform and known (Equation (7)); (b) no mass flow in the symmetry region (Equation (8)); (c) the convective boundary conditions on the grain surface are known (Equation (9)).

$$T = T_0 \text{ for } t = 0 \tag{4}$$

$$\frac{\partial T}{\partial t} = 0 \text{ for } r = 0 \tag{5}$$

$$\frac{\partial T}{\partial t} = k(T_w - T) \text{ for t} > 0 \tag{6}$$

$$c = c_0 \text{ for t} = 0 \tag{7}$$

$$\frac{\partial c_i}{\partial t} = 0 \text{ for r} = 0 \tag{8}$$

$$\frac{\partial c_i}{\partial t} = D(c_e - c) \text{ for t} > 0 \tag{9}$$

where: $T_0$ is the temperature of uncooked barley (K), $T_w$ is the water temperature (K), $k$ and $D$ are the heat transfer coefficient (W/m·K) and the mass transfer coefficient (m/s), respectively, t is the cooking time (s), and $c_0$ and $c_e$ are the initial moisture content (kg/kg) and the equilibrium moisture content (kg/kg) of the barley grains, respectively.

*2.4. Input Parameters*

The input parameters used to simulate the barley cooking process are shown in Table 1. The diffusion coefficient is discussed here in detail.

2.4.1. Diffusion Coefficient

Previous studies have proved the temperature dependency of the diffusion coefficient of grains under hydrothermal conditions, following an Arrhenius type relationship [14,15]. As a consequence of this equation (Equation (10)), the diffusion coefficient increases with temperature.

$$D_t = D_0 \exp\left(\frac{-E_a}{RT}\right) = 1.203 \times 10^{-5} \exp\left(\frac{-4147.7}{T}\right) \tag{10}$$

where $D_t$ is the effective coefficient of the mass transfer (m/s), $D_0$ is a constant, $E_a$ is the activation energy (J/mol), and R and T are the universal rate constant and absolute temperature (K), respectively. $D_0$ and $E_a$ are found to be equivalent to $1.203 \times 10^{-5}$ and 4147.7 J/mol, respectively [9].

**Table 1.** Input parameters used in the simulations for barley porridge cooking.

| Parameter | Value | Units | Source |
|---|---|---|---|
| Dimensions | | | |
| Major axis, a | 2.89 | mm | This study |
| Major axis, b | 1.59 | mm | This study |
| Major axis, c | 1.06 | mm | This study |
| Density | | | |
| Water, $\rho_w$ | 998 | Kg/m$^3$ | [17] |
| Barley, $\rho_b$ | 1304 | Kg/m$^3$ | [9] |
| Thermal conductivity | | | |
| Water, $k_w$ | $0.57109 + 0.0017625 - 6.7306 \times 10^{-6}T^2$ | W/m·K | [17] |
| Barley, $k_b$ | 0.1590 | W/m·K | [9] |
| Specific heat capacity | | | |
| Water, $C_{pw}$ | $4176.20 - 0.0909(T - 273) + 5.4731 \times 10^{-3}(T - 273)^2$ | J/kg·K | [17] |
| Barley, $C_{pb}$ | 1800 | J/kg·K | [9] |
| Equilibrium concentration of water, $c_e$ | 47,222 | Mol/m$^3$ | This study |
| Diffusion coefficient, $D$ | $1.203 \times 10^{-5} \exp(-4147.7/T)$ | m$^2$/s | [9] |
| Young's modulus, $E$ | Equation (11) | Pa | [18] |
| Poisson's ratio, $V_r$ | Equation (12) | – | [18] |
| Hygroscopic expansion coefficient of water, $\beta$ | $1.35 \times 10^{-3}$ | M$^3$/kg | This study |
| Molecular weight of water, $M_{mw}$ | 0.0180 | Kg/mol | |
| Initial conditions | | | |
| Water concentration, $C_0$ | 9598 | Mol/m$^3$ | This study |
| System temperature, $T_0$ | 298.15 | K | This study |

2.4.2. Mechanical Properties

Mechanical properties, such as the elastic modulus (*E*) and Poisson's ratio (*ν*) of barley, are required as functions of phase transition temperature ($T_g$). Barley starch granules are initially glassy,

and transform to a rubbery state when the temperature is higher than $T_g$. For barley starch in a glassy state, $E_g$ and $v_g$ have been reported as 500 MPa and 0.28, respectively [19]. In the rubbery state, the elastic modulus of starch ($E_r$) is expected to be of the order of 1 kPa [17]. Poisson's ratio in a rubbery state ($v_r$) has been estimated to be about 0.5. In order to avoid singularity and help with convergence of the numerical scheme, a value of 0.49 was adopted during computations. The temperature dependency of the elastic modulus and Poisson's ratio, in consideration of $T_g$, were approximated using the following functions [20]:

$$\mathrm{E(T)} = \frac{1}{2}\left(E_g + E_r\right) - \frac{1}{2}\left(E_g - E_r\right)\tanh\frac{T - T_g}{\beta} \tag{11}$$

$$v(\mathrm{T}) = \frac{1}{2}\left(v_g + v_r\right) - \frac{1}{2}\left(v_g - v_r\right)\tanh\frac{T - T_g}{\beta} \tag{12}$$

Here, $\beta$ is a parameter related to the temperature range across which phase transition occurs. A value of $\beta = 5\,^\circ\mathrm{C}$ was assumed based on values reported for other glassy polymers [18].

### 2.5. Solution Methodology Geometry, Mesh, and Implementation

A 3D geometry was used in this study and, owing to symmetry, a one-eighth ellipsoid was created (Figure 1). A tetrahedron mesh consisting of 9943 elements was used (Figure 2a), and the quality of the mesh was evaluated according to the color gradient (Figure 2b); the colors varied from red (low quality: 0) to green (high quality: 1).

A commercially available finite element software, COMSOL Multiphysics 5.3a (Comsol Inc., Burlington, MA, USA), was used to solve the governing equations described in Sections 2.1 and 2.2. The modules used were Transport of Diluted Species, Heat Transfer in Solids and Solid Mechanics for the mass transfer, heat transfer, and deformation phenomena, respectively. The simulation was calculated using a 2.93 GHz 6-core Intel Xeon Workstation with 24 GB RAM.

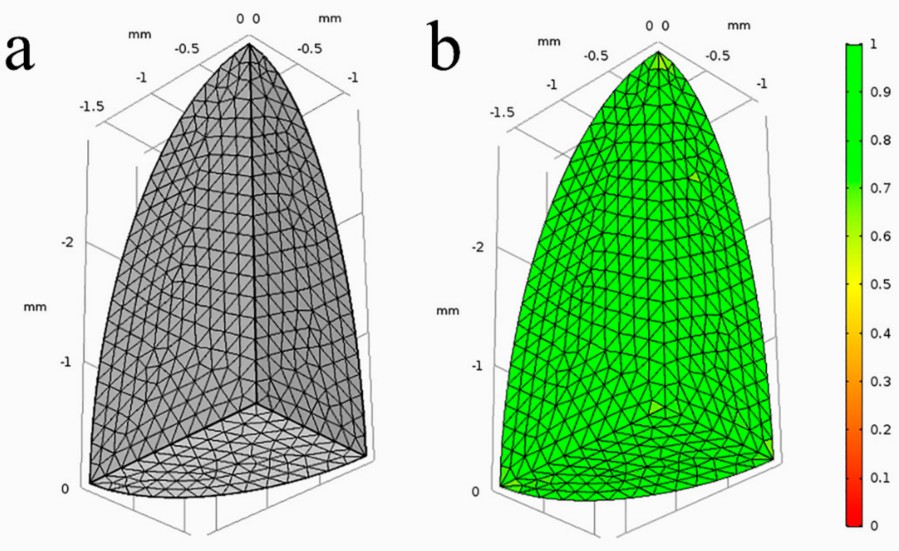

**Figure 2.** Meshed geometry with tetrahedron mesh elements (**a**); distribution of the evaluated quality of the mesh (**b**).

## 3. Experimental Methodology

### 3.1. Materials

Dehulled barley grains were obtained from Wal-Mart (China) Investment Co., Ltd. (Shenzhen, China). The barley grains were selected carefully so that the initial dimensions were almost the same.

The initial dimensions of the barley grains (a = 2.89 mm, b = 1.59 mm, c = 1.06 mm) were determined using a vernier caliper (precision: 0.01 mm).

### 3.2. Cooking Process

Barley porridge was prepared according to the method of Kwang et al. [21], with some modifications. Dehulled barley grains (15 g) were washed and rinsed three times with tap water. Water was added to the barley with the ratio of 8:1 (*v*/*w*) in a commercial tin can (ΦA = 73 mm, H = 59.8 mm) (ORG Packaging Co. Ltd., Beijing, China), which was then sealed using a hand seamer (YJ-C200, Zhangjiagang Yijie Automation Equipment Co., Ltd., Suzhou, China). The sealed can was put into an autoclave (Beijing Fanwen Trade Co. Ltd., Beijing, China) pre-heated to 60 °C. Then, the autoclave was further heated up to 100, 115, and 121 °C, maintained for 0–98.5, 0–86, and 0–79 min, respectively, and cooled down quickly using manual exhaust. In order to minimize the impact of the cooling stage on the changes in moisture and volume during porridge cooking, as soon as the vapor temperature dropped to around 100 °C, the can was transferred from the autoclave to cold water (25 °C) for further cooling until the water (inside the tin can) reached room temperature. The temperature changes of the vapor (heating medium, inside the autoclave chamber and outside the can) and water (heat transfer medium, inside the can) during cooking were monitored using MPIII Temperature Data Loggers (M4T12396, Mesa Laboratories, Inc., Lakewood, CA, USA). During the cooking periods at 100 °C and 121 °C, the vapor temperature changes and theoretical and actual time points for sample collection during cooking are shown in Figure 3.

Specifically, the vapor temperatures reached 100, 115, and 121 °C after 8.5, 16, and 19 min, respectively. In the heating-up period, the grains samples were collected at 0, 3, 6, and 8.5 min when the cooking temperature was 100 °C, 0, 3, 6, 8.5, and 16 min at 115 °C, and 0, 3, 6, 8.5, 16, and 19 min at 121 °C. At the stage of preservation, the collection interval was 5 min in the first 30 min, and thereafter, samples were collected every 10 min up to the equilibration of the hydration.

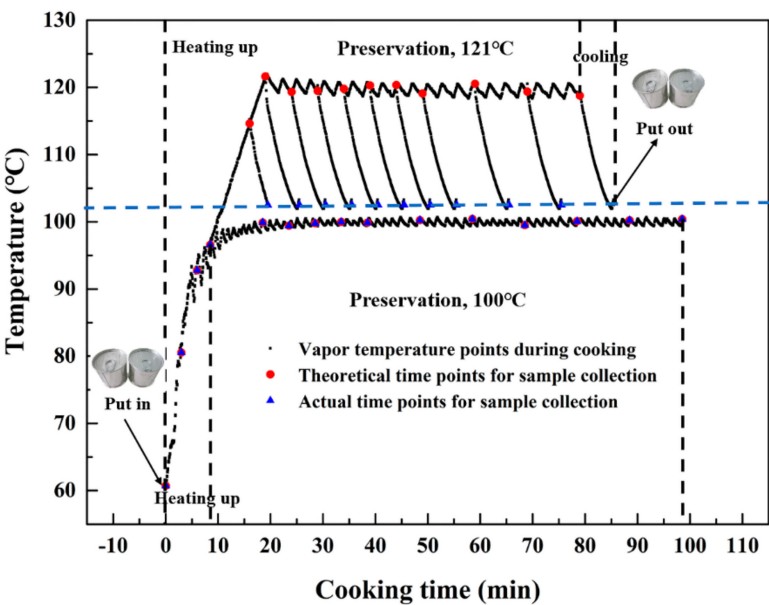

**Figure 3.** Temperature history and time points for sample collection (theoretical and actual) during cooking at 100 °C and 121 °C.

### 3.3. Measurement of Moisture Content and Volume Variation

After the removal of the surface water, the moisture content, volume, and expansion ratio of the cooked barley were analyzed. The moisture content (g·g$^{-1}$, wet basis) of the samples at each time step was obtained based on the increase in sample mass at the corresponding times [22]. The volume

variation (mm$^3$) and expansion ratio (mm$^3\cdot$mm$^{-3}$) were determined using water displacement in a measuring cylinder following the method of Fracasso et al. [23], with some modification. All the grains of the cooked barley were placed inside a 250 mL measuring cylinder, which originally contained 100 mL water. The total volume of 15 g cooked barley grains was equal to the volume increment of water. The expansion ratio of the cooked barley grains at different time points for collection was calculated by dividing the volume expansion at corresponding times with the volume of uncooked kernels. All the experiments described above were conducted in three replicates.

## 4. Results and Discussion

The model developed was validated by comparing the moisture content and expansion ratio change histories of the cooked barley. The performances of the models were determined according to their coefficient of determination ($R^2$), the root mean square error (RMSE, %), and the mean relative deviation modulus (P). Transient changes in the distribution of moisture content and the shape of the barley are also discussed next.

### 4.1. Moisture Absorption Characteristic

The moisture content changes of barley grains during the cooking process at 100 °C, 115 °C, and 121 °C are monitored and modeled. The plots of moisture content–cooking time are shown in Figure 4, and here the moisture content is represented by the increment of mass gain of barley grains. The experimental data fitted the simulated values well (Equations (1) and (2)), with the $R^2$ ranging from 0.993 to 0.997, RMSE ranging from 0.046 to 0.068 g$\cdot$g$^{-1}$, and P values below 5.058%, as shown in Table 2. According to Jideani and Mpotokwana [22], a P value of less than 10% indicates a good fit for practical purposes. However, the RMSE values in our present study are comparatively larger than the results reported by Perez et al. [16], wherein the RMSE values for the simulated water content of rice grains ranged from 0.0066 to 0.0252 g$\cdot$g$^{-1}$, in four hydration conditions. This is probably due to the different processing conditions used and the varieties of the kernels. Figure 4 also shows the hydration behavior of dehulled barley during different cooking conditions. In the heating-up period (I), the diffusion coefficient increases with temperature according to exponential law (Equation (10)). Hence, the hydration rate became increasingly fast although the water concentration gradient between the surface and the inside of the barley kernel was reduced simultaneously. At the stage of preservation (II), the curve exhibited the characteristic progression whereby an initial high rate of water gain is followed by slower absorption in a later stage. As cooking proceeds, the amount of moisture absorbed approaches an equilibrium value (about 3 g$\cdot$g$^{-1}$).

**Table 2.** Statistical analysis of the fitting of the numerical models to different moisture content and expansion ratio data of dehulled barley grains (100–121 °C) during canned porridge cooking.

| T (°C) | Moisture Content (g·g$^{-1}$) | | | Expansion Ratio | | |
|---|---|---|---|---|---|---|
| | $R^2$ | P (%) | RMSE (g·g$^{-1}$) | $R^2$ | P (%) | RMSE |
| 100 | 0.993 | 4.325 | 0.053 | 0.978 | 7.230 | 0.250 |
| 115 | 0.997 | 5.058 | 0.068 | 0.982 | 6.418 | 0.230 |
| 121 | 0.997 | 4.581 | 0.046 | 0.990 | 5.207 | 0.174 |

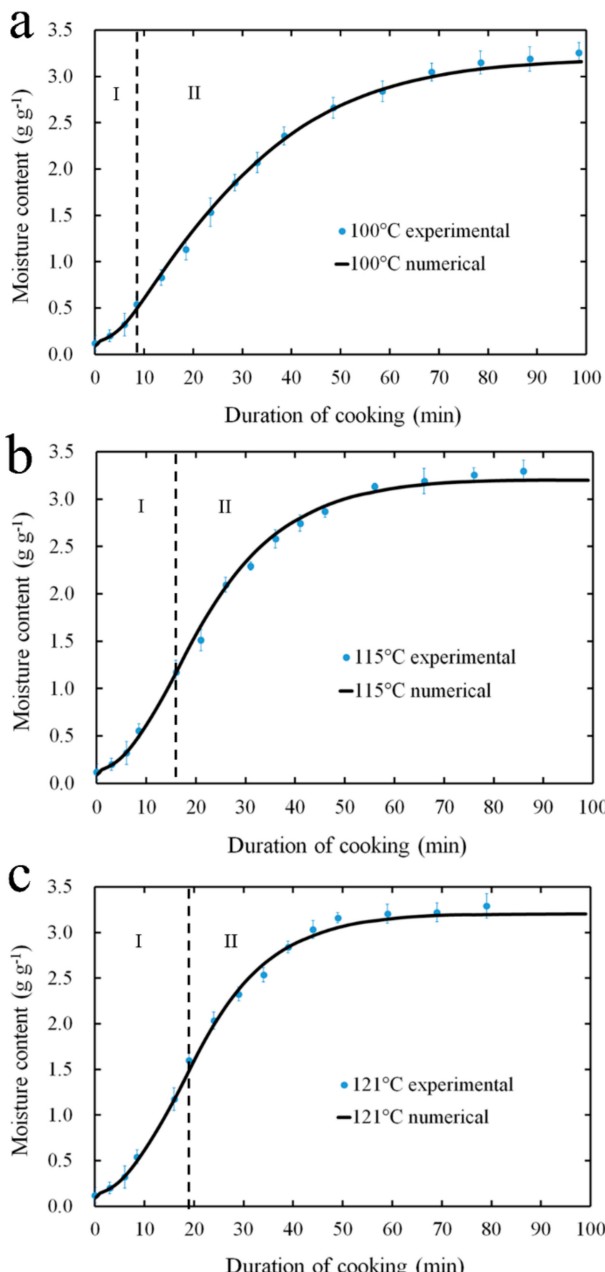

**Figure 4.** Changes in predicted and experimentally observed moisture content (wet basis) of barley kernels during cooking (I: heating-up stage, II: preservation) of canned porridge at 100 °C (**a**), 115 °C (**b**), and 121 °C (**c**). Vapor temperatures reached 100, 115, and 121 °C at 8.5, 16, and 19 min, respectively.

Figure 5 shows the evident change and the distribution of moisture in the dehulled barley during the cooking. Although the simulation of hydration was performed for all cooking temperatures, only the results for 121 °C are presented because these are representative of the other conditions explored. The different colors represent the different values of the moisture field. It can be observed that the surface layer of the grain is hydrated in the beginning of the cooking process, leaving the central core of the kernel dry. At 24 min of cooking at 121 °C, the average water content in barley is 1.989 g·g$^{-1}$, varying from to 1.2630 to 2.2957 g·g$^{-1}$ according to the position inside the kernel (Figure 5). After 49 min of cooking, the average moisture content is 3.205 g·g$^{-1}$ (Figure 4), uniformly distributed across the kernel. This indicated that the equilibrium value was reached. A similar moisture diffusion performance was reported by Montanuci et al. [9], when the barley grain was soaked at a temperature of 10–25 °C.

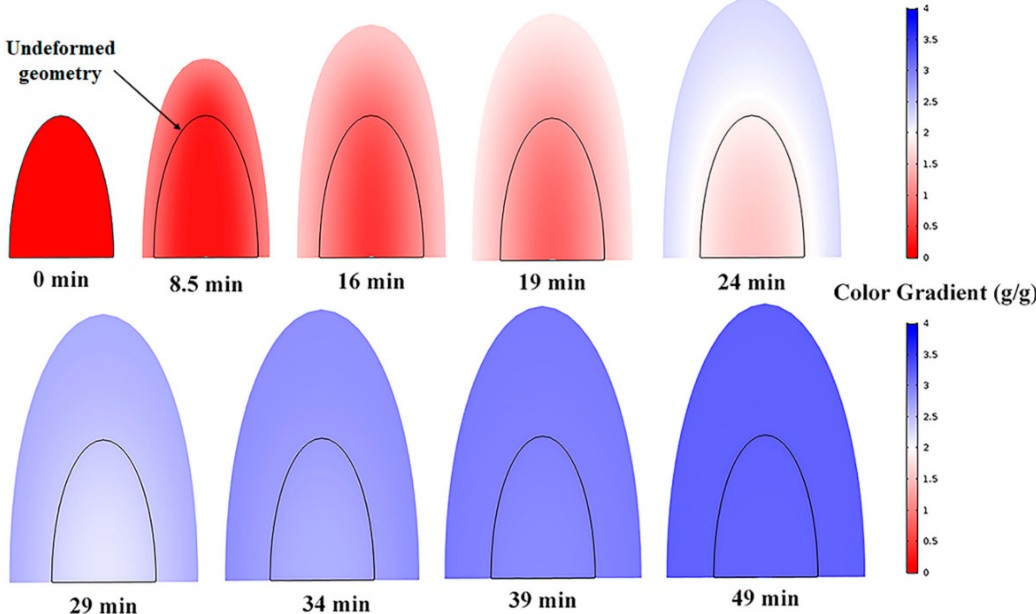

**Figure 5.** Simulation of moisture uptake in the cooking process at 121 °C.

*4.2. Volume Change During Hygroscopic Swelling*

The volumetric changes of barley kernels cooked at three different temperatures in sealed cans are presented in Figure 6. The barley kernels swelled obviously throughout the geometry as soon as the grains began to absorb water. The volumes of the barley samples expanded faster at a higher temperature. It took about 49 min to reach the maximum expansion at 121 °C, whereas 78.5 min was required at 100 °C. This observation was consistent with the results reported by Amogha et al. [24], wherein the equilibrium length of rice grain was achieved in 60, 30, and 20 min, respectively, for pre-soaked rice, and 70, 40, and 35 min respectively for un-soaked rice when the temperature was 80, 90, and 97 °C. This could be explained by the fact that water diffusion occurs more readily at higher temperatures [25]. The binding of starch molecules and water induces the volume expansion of barley grains [26].

The experimental data relating to the volume of the grain can also be reasonably fitted using the simulated values (Equation (3)). In our study of volume simulation, the $R^2$ for the simulated volumes of dehulled barley were 0.978, 0.982, and 0.990, the P values were 7.230%, 6.418%, and 5.207%, and the RMSE values were 0.250, 0.230, and 0.174, respectively, for 100, 115, and 121 °C (Table 2). The RMSE values are relatively close to those in the report of Perez et al. [15], which range from 0.184 to 0.794. Therefore, the numerical model can be used in the prediction of the swelling of barley. The results of the simulation of the hygroscopic swelling in barley grains using the fixed value of the hygroscopic expansion coefficient were a little bit off at 115 °C and 121 °C (Figure 6).

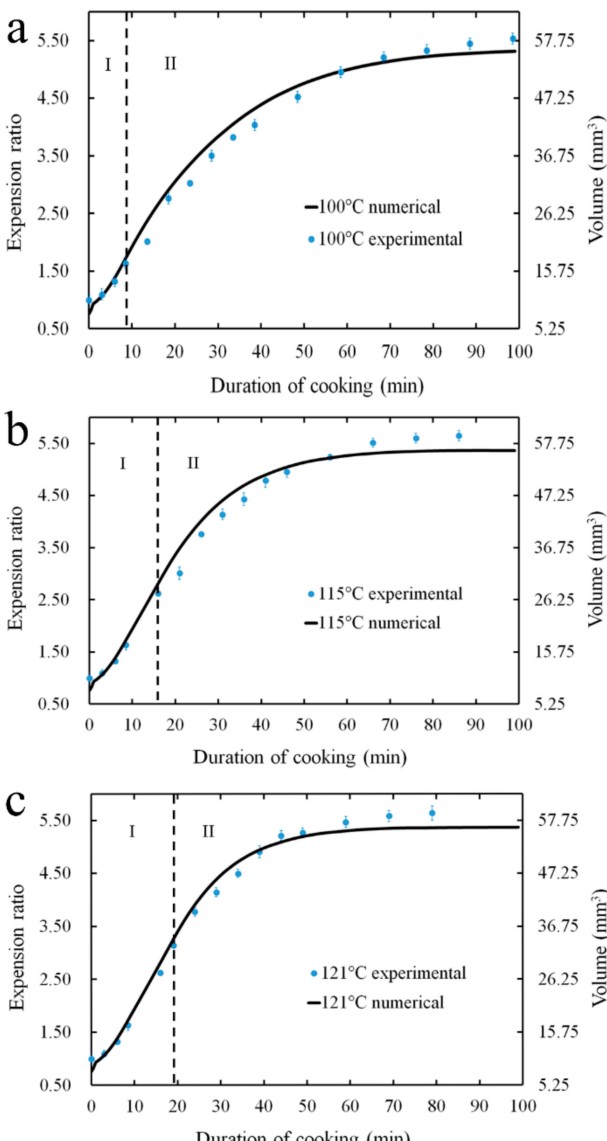

**Figure 6.** Computed and measured changes in volume and expansion ratio of barley kernels during cooking (I: heating-up stage, II: preservation) of canned porridge at 100 °C (**a**), 115 °C (**b**) and 121 °C (**c**). The vapor temperatures reached 100, 115, and 121 °C at 8.5, 16, and 19 min, respectively.

As mentioned in Section 4.1, the results at 121 °C are representative of other conditions. Both the experiment and the simulation dynamics of the swelling characteristics of barley at 121 °C are shown in Figure 7. The variation in the shade of the kernel shows obvious volumetric changes in barley. The volumetric expansion of the barley begins soon after the water has been absorbed throughout the kernels. As is shown in Figures 6 and 7, the swelling is more pronounced after 8.5 min of cooking (the expansion ratio is only 1.64 at 8.5 min, but 2.62 at 16 min), and becomes less obvious after 49 min (the volume increment is only 3.9 mm when the cooking time increases from 49 min to 79 min). It is obvious that the swelling of the kernels was higher at the tip than along the sides of the kernels during cooking. The major reason for this observation is the geometry of the barley kernels. On account of the elliptical shape of the kernel, more starch molecules are located along the sides of the kernel than at its tip [16]. The swelling of the starch granules along the side section resulted in greater compressive stress. This force pushes more starch molecules towards the tip of the kernel, causing more starch granules to further elongate.

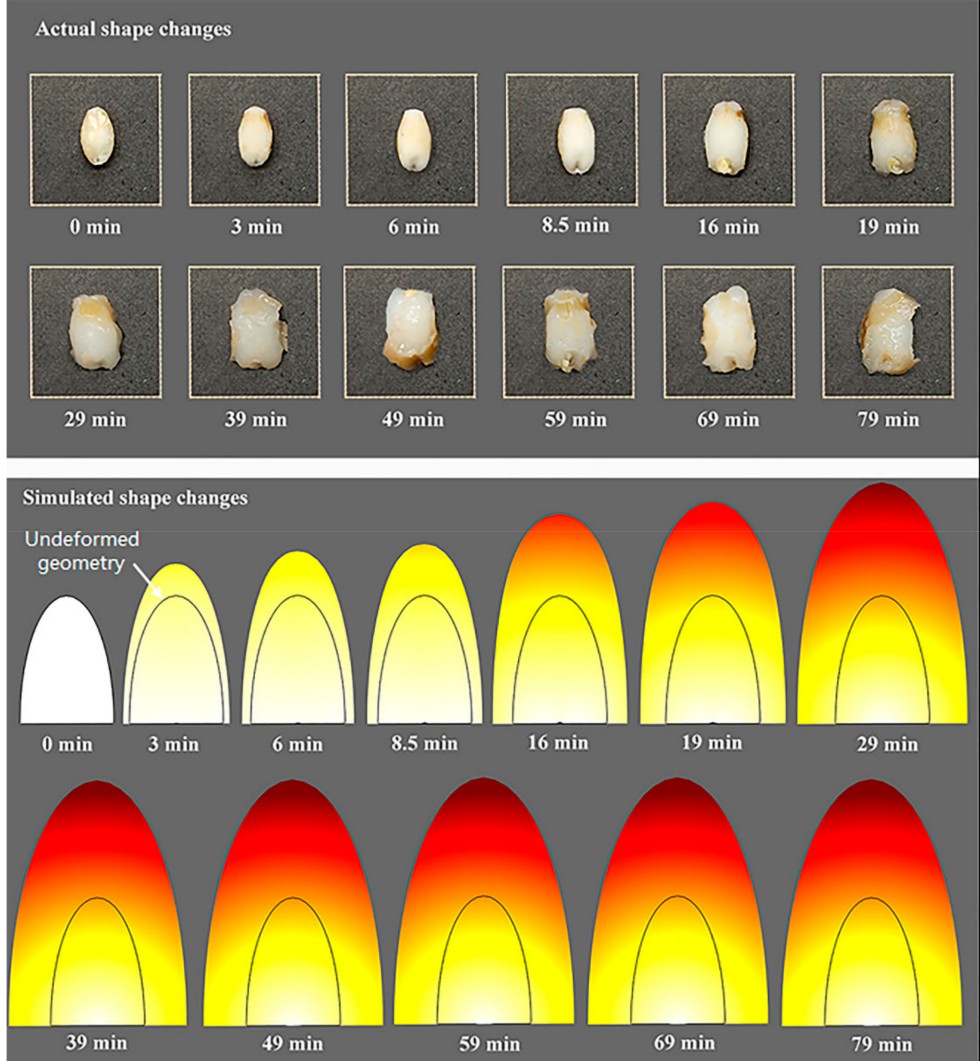

**Figure 7.** Computed and experimentally observed shape changes of barley kernels at different cooking times (121 °C).

## 5. Conclusions

Cooking canned barley porridge is a complicated process, which involves several processes, such as moisture uptake, gelatinization of starch, and swelling taking place simultaneously. Computational techniques and numerical modeling can be used to study and understand these complex changes occurring during the cooking of canned porridge, which helps to optimize the process with consequences for the curve of the moisture content and its distribution.

In the current study, a numerical simulation was developed using the finite element method making it possible to predict the amount and distribution of moisture content, as well as the volume of barley during the cooking of canned porridge. The phenomenological model, verified by experiments, can be applied to predict and optimize the hydrothermal processes of barley and other cereals, even in conditions not tested experimentally. The rate of hydration and swelling increased evidently with temperature increments, leading to a decrement of about 50% of the time to reach the equilibrium moisture content and volume of barley grains by increasing the cooking temperature from 100 to 121 °C. However, temperature does not noticeably affect the equilibrium value of the average moisture content.

**Author Contributions:** D.L. put forward the idea of this work, L.W. conducted the simulation and wrote this paper, M.W. contributed to the results analysis and post-processing, M.G. revised this paper, T.D. and X.Y. supervised the process.

**Acknowledgments:** This research was funded by the National Major R & D Program of China (grant no. 2016YFD0400301).

**Conflicts of Interest:** The authors declare that no conflicts of interest exist.

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
