# Peer review of "Numerical Simulation of Water Absorption and Swelling in Dehulled Barley Grains during Canned Porridge Cooking"

_processes, doi:10.3390/pr6110230_

Reviewer 1 Report

To Authors,

the manuscript (ref Processes-382442) titled “Numerical simulation on water absorption and swelling in dehulled barley grains during cooking of canned porridge” by Wang et al. studied the hydration behavior of dehulled barley grain during cooking of barley porridge in a sealed tin can at 100, 115 and 121°C. Authors developed a 3D numerical model to describe water uptake and hygroscopic swelling. “The proposed model was found to be adequate”.

Seemingly, the paper addresses a key issue in the industrial processing of (dehulled) barley porridge and potentially be of use in the case of other cereals and conditions. A cursory online search in bibliographic databases,  attests it is a current topic of research. The paper is within the scope and follows the format and style of Processes but its English requires revision in a few instances. In the Introduction, Authors contextualize the study well and define the objective appropriately. They should provide a few details about the numerical model development and the experimental methodology carried out in the Model development and Experimental methodology sections. In the Results and Discussion, Authors are required to complete their reasoning and/or discuss their findings in light of published research (e.g. Briffaz, Aurélien, et al. "Modelling of water transport and swelling associated with starch gelatinization during rice cooking." Journal of Food Engineering 121 (2014): 143-151. Shinde, Yogesh H., et al. "Design and development of energy efficient continuous cooking system." Journal of Food Engineering 168 (2016): 231-239. Jian, Fuji, et al. "Water sorption and cooking time of red kidney beans (Phaseolus vulgaris L.): part II–mathematical models of water sorption." International Journal of Food Science & Technology 52.11 (2017): 2412-2421. Shinde, Yogesh H., et al. "Kinetics of cooking of unsoaked and presoaked split peas (Cajanus cajan)." Journal of Food Process Engineering 40.5 (2017): e12527. Amogha, V., et al. "Image analysis based validation and kinetic parameter estimation of rice cooking." Journal of Food Process Engineering 40.5 (2017): e12552.). In the Conclusion section, the final statements on significance of results were not in fact tested statistically. Most of the refs cited are rather recent (post-year 2010).

Specific comments and suggestions are made directly in the PDF of the submitted manuscript using tools available in Adobe Acrobat Reader DC. Surely, the Authors will be able to consider them.

Author Response

Response to Reviewer 1 Comments

Point 1: The author should clarify the cooking of barley is industrial or home (line 39)

Response 7: Thanks for your kind suggestion. The “Cooking of barley” has been revised to be “Cooking of barley porridge in industry”. (line 41)

Point 2: Can Authors provide evidence(s) of appropriateness of this geometry to describe barley grain (dehulled), e.g. scientific illustration? (line 95)

Response 28: Yes, we can. The shape of the barley grain has been simplified and considered to be an ellipsoid by Montanuci et al. (2014). And the rice grain was usually simplified to be an ellipse (Gulati and Datta, 2016; Amogha et al., 2017), whose geometry is similar to barley.

Cited from:

Montanuci, F. D., Perussello, C. A., de Matos Jorge, L. M. & Jorge, R. M. M. (2014). Experimental analysis and finite element simulation of the hydration process of barley grains. Journal of Food Engineering, 131), 44-49.

Amogha, V., Shinde, Y. H., Pandit, A. B. & Joshi, J. B. (2017). Image analysis based validation and kinetic parameter estimation of rice cooking. Journal of Food Process Engineering, 40(5), e12552.

Gulati, T. & Datta, A. K. (2016). Coupled multiphase transport, large deformation and phase transition during rice puffing. Chemical Engineering Science, 139(75-98).

Point 3: Consider reformatting Table 1 to fit a "portrait" orientation that reads better. Check ref. format! What do [number] mean? Why the 2 formats, (Author, year) and [number]?

 (line 124)

Response 3: Thanks for your kind suggestion. We have reformatted Table 1 to fit a “portrait” orientation. And we normalizeds the ref. format. (line 125)

Point 4: What were the criterion/criteria for numerical optimization used to find the solution? (line 145-146)

Response 4: The aim of our study is to build a 3D numerical model to describe the changes of moisture content and volume of barley grains during the cooking process of canned porridge according to the Fick’s second Law, Fourier’s Law, Hookean Law, physical parameters of barley grains and the boundary and initial conditions. This model can’t give an optimized solution directly, but can be used to evaluate the effect of processing time and temperature on the moisture absorption and volume expansion. That is, this model can provide the minimum time for the barley to reach the required moisture content and volume. In our study, the required value is considered as equilibrium value.

Point 5: Clarify the precision of the vernier caliper used. (line 155)

Response 5: Thanks for your kind suggestion. We have clarified the precision of the vernier caliper in our study. It’s 0.01 mm. (line 157)

Point 6: Clarify the material of the can used (line 159)

Response 6: Thanks for your kind suggestion. We have clarified the material of the can used in our study. It’s tin. (line 161)

Point 7: Clarify the temperature of the cold water (line 166)

Response 7: Thanks for your kind suggestion. We have clarified the temperature of the cold water in our paper. It’s 25°C. (line 168)

Point 8: “……the can was transferred from the autoclave to the cold water for further cooling until reached room temperature” Clarify which part of the canned porridge reaching room temperature (line 167)

Response 8: Thanks for your kind suggestion. It’s the water (inside the tin can) reaching room temperature. 

The sentence has been revised to: “……the can was transferred from the autoclave to cold water (25°C) for further cooling until water (inside the tin can) reached room temperature” (line 169)

Point 9: Out of curiosity, how was sampling carried out in practice?

Response 9: Take the cooking condition (121°C, 49min) as an example: unbroken dehulled barley grains (15 g, random sampling) were put into a can, then the can was put into the per-heated autoclave (60°C), went through the stage of heating up (19 min), preservation (40min), and cooling to 100°C (Ignore). After that, the can were transferred from the autoclave into cold water (25°C) for further cooling until the water (inside the tin can) reached room temperature. Finally, all the barley grains were collected from the tin can for the next measurement.

Point 10: Clarify the simulated values of moisture content obtaining from which equation(s). (line 197)

Response 10: Thanks for your kind suggestion. The simulated values were obtained from (Eq. (1) and (2)). (line 197)

Point 11: Consider rewriting this sentence, “Starting from the lowest cooking temperature, the values of the R2 were 0.993, 0.997 and 0.997, RMSE ……and 4.581%”, also, copy info to plots in figure or its caption. (line 199-201)

Response 11: Thanks for your kind suggestion. The sentence has been rewritten to “The experimental data fitted well the simulated values, with the R2 ranging from 0.993 to 0.997, RMSE ranging from 0.046 g g-1 to 0.068 g g-1 and P values below 5.058%, as shown in Table 2”. (line 205-206)

And I created a table for R2, RMSE and P of predicted and experimentally observed moisture content and simulated volume. (line 220)

Point 12: Put the error in the experimental curves. (line 213)

Response 12Thanks for your kind suggestion. We redrew the figure and put the error in it. (line 224)

Point 13: The numerical/model prediction line results from which eq(s)? (line 213)

Response 13We can get the relationship between instantaneous water content (c) and cooking time (t) from (Eq. (1) and (2)).  (line 224)

Point 14: Seemingly this line depicts initial size? State in caption below. (line 218)

Response 14: Thanks for your kind suggestion. We redrew the Fig. 5 and added a note “Undeformed geometry” to tell readers that the line depicts initial size.  (line 230)

Point 15: Why only the results for 121°C are presented? (line 222)

Response 15 Because these results are representative of other conditions explored. (line 234 -235)

Point 16Discussion? Comparison with (published) literature?  (line 228)

Response 16: Thank you for your gracious advice. I have added some discussions and references into section 4.1. For example, “Similar moisture diffusion performance has been reported by Montanuci et al. [9], when the barley grain is soaked at the temperature of 10 - 25°C.” (line 240 - 242)

Point 17Discussion? Comparison with (published) literature?  (line 237)

Check e.g.

1. Briffaz, Aurélien, et al. "Modelling of water transport and swelling associated with starch gelatinization during rice cooking." Journal of Food Engineering 121 (2014): 143-151.
2. Shinde,Yogesh H., et al. "Design and development of energy efficient continuous cooking system." Journal of Food Engineering 168 (2016): 231-239.
3. Jian, Fuji, et al. "Water sorption and cooking time of red kidney beans (Phaseolus vulgaris L.): part II–mathematical models of water sorption." International Journal of Food Science & Technology 52.11 (2017): 2412-2421.
4. Shinde, Yogesh H., et al. "Kinetics of cooking of unsoaked and presoaked split peas (Cajanus cajan)." Journal of Food Process Engineering 40.5 (2017): e12527.
5. Amogha, V., et al. "Image analysis based validation and kinetic parameter estimation of rice cooking." Journal of Food Process Engineering 40.5 (2017): e12552

Response 17Thank you for your gracious advice and valuable information. I have carefully read the materials given by you, and added some discussions and references into section 4.2. For instance, “This observation was consistent with the results reported by Amogha et al. [26] that the equilibrium length of rice grain was achieved in 60, 30 and 20 min respectively for pre-soaked rice and 70, 40 and 35 min respectively for un-soaked rice when the temperature was 80, 90 and 97°C. It could be explained that the water diffusion occurs more readily at higher temperature [27]. The binding of starch molecules and water induces the volume expansion of barley grains [28].”   (line 249-253)

Point 18: Model line does not fit the data points as well as in Fig 5 for example. Analysis of residuals would help discern if, for instance, model is ill-fitted or inappropriate for the data at hand? (line 238)

Response 18: Because the swelling of barley grains was not isotropy actually, so the model does not fit the data points as well as in Fig. 5. But it can be also ued to predict the volume expansion well.

Point 19: The numerical/model prediction line results from which eq(s)? (line 238)

Response 19We can get the relationship between volume expansion and cooking time from (Eq. (3)), these data points consist of the model prediction line.  (line 254)

Point 20: Why only the results for 121°C are presented? (line 254)

Response 20As mentioned in Section 4.1, these results are representative of other conditions. (line 266)

Point 21: Why the swelling was more pronounced after 8.5 min of cooking? (line 255) Don’t agree the swelling became unnoticeable at 49 min (line 256)

Response 21As is shown in Fig. 6c and Fig. 7, the swelling is more pronounced after 8.5 min of cooking (the expansion ratio is merely 1.64 at 8.5 min, but 2.62 at 16 min), and becomes less obvious after 49 min (the volume increment is only 3.9 mm when cooking time increases from 49 min to 79 min).   (270-272)

Point 22: “On account to an ellipse shape of the kernel, more starch molecules are locating along the sides of the kernel than at the tip of the kernel” Citation needed here. (line 258-259)

Response 22Thanks for your kind suggestion. We add a citation here. (276)

Point 23: There are many grammar, tense and format mistakes in the manuscript.

Response 23: Thanks for your kind advices. We have checked the grammar, tense and format mistakes throughout our manuscript and made corrections carefully. All the corrections were highlighted in yellow in the revised manuscript and the details were listed as follows

1 on (line 2) →of (line 2)

2 The word “that” (line 17) has been deleted (line 19)

3 ranged (line 17) →ranging (line 19)

4 value (line 20) →values (line 22)

5 cereal (line 21) →cereals grains (line 23)

6 several (line 34) →few (line 36)

7 were (line 40) →are (line 42)

8 kernel (line 43) →kernels (line 48)

9 assimilated (line 44) →assimilation (line 49)

10 The word “just” (line 48) has been deleted (line 53)

11 The word group “in a study” (line 52) has been deleted (line 57)

12 his (line 53)→their (line 58)

13 build (line 57)→built (line 62)

14 The word “the” (line 58) has been deleted (line 63)

15 The word “large” (line 64) has been deleted (line 68)

16 The word “required” has been added (line 73)

17 was (line 72)→is (line 75)

18 in (line 74)→at (line 77)

19 were (line 75)→are (line 78)

20 were (line 76)→is (line 79)

21 were (line 77)→is (line 80)

22 The word “Law” has been added (line 83)

23 Laws (line 81)→Law (line 84)

24 is (line 88)→are (line 90)

25 is (line 89)→are (line 91); swells (line 89)→swell (line 91)

26 Thanks for your kind suggestion. We followed your advice and chose the term “strain” (line 90). (line 92)

27 molecular (line 91)→molecules’ (line 93)

28 was (line 98)→is (line 99); was (line 98)→is  (line 99)

29 was (line 105)→are (line 106)

30 occurred (line 106)→occurs (line 107)

31 was (line 109)→are (line 110)

32 was follow (line 117)→following  (line 120)

33 The title “2.4.1 Diffusion coefficient” was added (line 118)

34 were (line 117)→are (line 124)

35 was (line 139)→is (line141); was (line 140)→is (line 142); was (line 141)→is (line 143); was (line 144)→is (line 146); were (line 145)→are (line 147); was (line 146)→is (line 148)

36 by (line 160)→using (line 163)

37 “when autoclave was” was deleted (line 165)

38 “the” was deleted (line 169)

39 at (line 173)→after (line 176)

40 are fitted together with (line 173)→fitted well (line 205)

41 “the previous report of” was deleted (line 202)

42 is a little bit (line 204)→are comparatively (line 208)

43 than results in the previous study (line 204)→probably (line 210)

44 “by” was added (line 216)

45 “4” (line 211) →“3” (line 217)

46 significant (line 220)→evident (line 232)

47 “and all the regions are occupied by navy blue” (line 227) →“uniformly distributed across the kernel”  (line 239-240)

48 vale (line 228)→value (line 239-240)

49 imbibe (line 232)→absorb (line 247)

50 the sentence (line 243-244) was deleted

51 “reasonably” was added (line 258); together with (line 242)→by the (line 258)

52 “respectively for 100°C, 115°C and 121°C (Table 2)” was added (line 261)

53 courses (line 252)→dynamics (line 267)

54 significant (line 253) →obvious (line 268)

55 “as the” (line 266) →that involves (line 283)

56 line 271: “was developed using finite element method” was added (line 288)

57 This (line 273) →The (290)

58 significantly (line 276) →obviously (line 293)

59 insignificantly (line 279) → unnoticeably (line 295)

Thank you for all your constructive advice.

Reviewer 2 Report

Why the author chose thhese three specific temperatures: 100, 115 and 121°C?

Introduction

line 36: please explain deeply why the ready-to-eat (RTE) porridge has excellent storage stability

line 40: explain the cooking conditions.

Lines 67-70: this sentence seems a result not an introduction, please modify.

2.4.1 mechanical properties

line 129: explain Eg and νg

Experimental  methodology

line 155:please write the initial experimental dimensions of barley grains

Fig. 3. in this figure why the authors did not put the temperature history and time points for sample collection (theoretical and actual) during cooking at 115°C?.

Line 182: the authors wrote: volume variation (mm3) and expansion ratio (mm3 182 mm-3) were determined by water displacement in a cylinder. Did they measure this variation at the end of cooking? please explain the procedure.

Figure 4: put the error in the experimental curves

I suggest to the authors to create tables for R2, RMSE of predicted and experimentally observed moisture content and simulated volume.

Why the authors did not measure Tg experimentally?

Author Response

Response to Reviewer 2 Comments

Point 1: Why the author chose these three specific temperatures: 100, 115 and 121°C?

Response 1: The cooking process of canned barley porridge is the sterilization process at the same time. We choose three common sterilization temperatures, 100, 115 and 121°C.

Point 2: please explain deeply why the ready-to-eat (RTE) porridge has excellent storage stability. (line 36)

Response 2: The cooked grains of common commercial RTE porridge were usually subjected to hot air drying till desired moisture content (~8%), and then be packaged in sterile conditions, so the RTE porridge has excellent storage stability.

Point 3: explain the cooking conditions. (line 40)

Response 3: Thanks for your suggestion. We have explained the cooking conditions in our study. (line 42-43)

Point 4: this sentence seems a result not an introduction, please modify. (line 67-70)

Response 4: Thanks for your suggestion. We have rewritten this sentence. (line 71-73)

Point 5: explain Eg and νg (line 129)

Response 5: Eg is the elastic modulus of barley starch in glassy state. νg is the Poisson’s ratio of barley starch in glassy state.

Point 6: please write the initial experimental dimensions of barley grains (line 155)

Response 6: Thanks for your suggestion. We have added the initial dimensions of barley grains. (line 157)

Point 7: Fig. 3. Why the authors did not put the temperature history and time points for sample collection (theoretical and actual) during cooking at 115°C.

Response 7: Because results at 121°C are representative of other conditions, so we take the results at 121°C for instance. In Fig. 3, we just put the temperature history and time points for sample collection during cooking at 121°C. In order to exhibit the cooling stage better, we put the temperature at 100°C as a reference.

Point 8: the authors wrote: volume variation (mm3) and expansion ratio (mm3 mm-3) were determined by water displacement in a cylinder. Did they measure this variation at the end of cooking? please explain the procedure.  (line 182)

Response 8: Thanks for your suggestion. We have explained the procedure in detail. (line 185-186, 189-194)

Point 9: put the error in the experimental curves

Response 9: Thanks for your suggestion. We redraw the figures and put the error in the experimental curves. (line 224 and line 254)

Point 10: I suggest to the authors to create tables for R2, RMSE of predicted and experimentally observed moisture content and simulated volume

Response 10: Thanks for your suggestion. We have created the Table 2. (line 220-223)

Point 11: Why the authors did not measure Tg experimentally?

Response 11: In my opinion, the model should be built based on the existing theories and physical properties firstly, and then validated by experimental data. If the suitable values of physical properties of barley grains can’t be obtained, we should measure the value experimentally.

Thank you for all your constructive advice.

Reviewer 3 Report

The article deals with the modeling and the respective experimental validation of the most important phenomena occurring during cooking of canned porridge, i.e. water absorption and swelling of barley grains.

The topic is original, interesting, and relevant to a broad audience covering academia and industry. Both model and experiments are well designed, and results are good to encouraging, conveying useful information and potentially stimulating further research.

However, the Abstract misses emphasizing the experimental part of the study, which is very important, few important statements throughout the text are unsupported, and the phrasing is sometimes confused. The language needs some changes.

Trying to help the authors improve the clarity of presentation, readability and attractiveness for readers, I've posted many comments, which are available in the herein enclsoed document.

I suggest publication of the manuscript, but only after the authors have considered all my comments and reacted accordingly.

Author Response

Response to Reviewer 3 Comments

Point 1: More emphasis should be given to the model validation by experiments carried out in the frame of this study. Based on the abstract, the reader could think that the model was validated on the basis of literature data.  (line 12-21)

Response 1: Thanks for your kind suggestion. In order to emphasize the part of model validation by experiments in the frame of this study, we rewrote the abstract. The corrections were highlighted in blue. (line 13-18)

Point 2: Explain why water absorption is so important.

Response 2: Changes in moisture content and volume of barley grains are two main phenomena during cooking. The control of moisture content in barley grains during cooking is of great importance as water molecules paly many parts in food reactions and food quality. In order to predict the optimum processing conditions, the water content variations with the grains are needed as a quantitative information. (line 43-47)

Point 3: Invert: mass transfer first, then heat transfer. (line 80-82)

Response 3: Thanks for your kind suggestion. We have changed the order of mass transfer and heat transfer. (line 82)

Point 4: Use a reference (line 92)

Response 4: Thanks for your kind suggestion. The reference 16 was inserted. (line 94)

Point 5: In order to ease interpretation, specify that, as a consequence of this equation, the diffusion coefficient increases with temperature. (line 118)

Response 5: Thanks for your kind suggestion. We rewrote this sentence as “As a consequence of this equation (Eq. (10)), the diffusion coefficient increases with temperature.” (line 120-121)

Point 6: Explain the criteria according to which the quality was assessed. (line 141)

Response 6It’s according to whether it helps with convergence of the numerical scheme or not.

The detail reason can be read here:

https://wenku.baidu.com/view/c92359e54bfe04a1b0717fd5360cba1aa8118c28.html

Point 7: What are "theoretical" and "actual" for? Wouldn't it be enough to show actual sampling times? (line 171)

Response 7Take the condition (121°C, 49 min) for instance, the theoretical time for sample collection is at 49min, but the actual time is longer than it because the tin can only can be transferred from the autoclave after the cooling stage. In Fig. 3, the red points are the theoretical time points for sample collection, the blue points are the actual time points after cooling stage.

Point 8: Based on Fig. 4, the equilibrium value looks like to be slightly greater than 3 g/g. (line 212)

Response 8Thanks for your kind suggestion. I have corrected it. (line 27)

Point 9: please reword distribution of water molecular(line 212)

Response 9Thanks for your kind suggestion. I have rewritten it. (line 287)

Point 10: There are many grammar, tense and format mistakes in the manuscript.

Response 10: Thanks for your kind advices. We have checked the grammar, tense and format mistakes throughout our manuscript and made corrections carefully. All the corrections were highlighted in blue in the revised manuscript and the details were listed as follows

1 The (line 12)Understanding the (line 12)

2 line 12-13, “because it is an important phenomenon for optimizing” has been deleted

3 assimilated (line 44)assimilation (line 49)

4 temperature (line 57) temperatures (line 61)

5 build (line 57) built (line 62)

6 evaluate (line 65) evaluate (line 70)

7 P (line 86) ρ (line 87)

8 is (line 88) are (line 90)

9 absorbs (line 89) absorb (line 91); swells (line 89) swell (line 91)

10 the hygroscopic swelling of starch molecular (line 91) starch molecules’ swelling (line 93)

11 was follow (line 117) following (line 120)

12 were (line 121) are (line 124)

13 was (line 158) were (line 161)

14 Take (line 170) During (line 173)

15 were (line 172) are (line 174)

16 closed (line 246) close (line 262)

17 representing (line 277) →leading to (line 293)

Thank you for all your constructive advice.

Round  2

Reviewer 1 Report

To Editor/Authors,

In the revised version of manuscript (ref Processes-382442-rd2) titled “Numerical simulation of water absorption and swelling in dehulled barley grains during cooking of canned porridge” by Wang et al., Authors addressed all comments and suggestions made to the original submission incl. correction of English, changes in figures and tables, completion of assertions with citations and expansion of the discussion of their findings in light of published literature by citing a few more refs. At this time, the manuscript is acceptable for publication in Processes.

Reviewer 2 Report

The authors improved the quality of paper. I suggest to accept in this new version the paper.